# HEADS UP: Design and Methods of a Louisiana State-Funded Surgical and Non-Surgical Weight Loss Program

**DOI:** 10.3390/ijerph17092999

**Published:** 2020-04-26

**Authors:** Phillip J. Brantley, Win Guan, Ricky Brock, Dachuan Zhang, Gang Hu

**Affiliations:** Pennington Biomedical Research Center, 6400 Perkins Road, Baton Rouge, LA 70808-4124, USA

**Keywords:** weight loss, bariatric surgery, intensive medical intervention, obesity, observation study

## Abstract

This paper describes the methodology, design and procedures used in the HEADS UP Project, an observational study to examine the feasibility of a state-funded weight loss program. HEADS UP offered two weight loss approaches: bariatric surgery or a non-surgical intervention composed of medical management, a low-calorie liquid diet and lifestyle change promotion. Participants were recruited through a multi-stage screening process, in-person interviews, and an initial low-calorie diet program. Eligible participants were entered into a lottery system, with 100 participants selected for the surgical group and 200 selected for the non-surgical group annually for five years. Anthropometric, clinical, and psychosocial assessments were completed at baseline and follow-ups. More than 6800 individuals completed the initial web screening. Screening procedures yielded 1412 participants (490 surgical and 922 non-surgical). Approximately 84% of the total participant population were female and 38% were Black. Participants had an average body mass index of 47.9 and 43 kg/m^2^ in the surgical and non-surgical groups, respectively. Recruitment and enrollment results of the HEADS UP study demonstrated significant interest in both the surgical and non-surgical treatment programs for obesity. These results support the feasibility of providing a state-funded weight loss program within a healthcare setting.

## 1. Background

National-level data (National Health and Nutrition Examination Survey) have indicated that more than one-third (37.7%) of American adults are classified as obese (body mass index [BMI] ≥30 kg/m^2^) and 7.7% are classified as extremely obese (BMI ≥40 kg/m^2^) [1]. More concerning, state-level data (Behavioral Risk Factor Surveillance System) have shown that Louisiana has the highest obesity rate in America [2]. This is a significant public health concern given that the obesity epidemic is strongly associated with the parallel epidemics of type 2 diabetes and other associated comorbidities (e.g., cardiovascular disease, hypertension, dyslipidemia and cancer). Despite the growing evidence demonstrating the medical efficacy and cost effectiveness of obesity treatments, insurance providers are still reluctant to provide financial reimbursements. 

Clinical practices have shown positive weight loss results in previous clinical efficacy research [3,4,5]. Randomized controlled trials [6,7,8,9,10] and observational studies [11,12] have also shown that bariatric surgery can produce significant short- and long-term weight loss in individuals with obesity. Studies have shown that bariatric surgery can be effective in treating chronic diseases closely associated with obesity such as type 2 diabetes [10], hypertension [4,8,13], and hyperlipidemia [4,8,13]. 

In 2011, the Office of Group Benefits of the State of Louisiana (OGB) partnered with the LSU Pennington Biomedical Research Center to direct the HEADS UP study, a prospective, observational study funded by OGB. This project was a product of House Resolution HCR55 requiring OGB to examine the feasibility and potential financial health benefits of providing obesity treatments to its adult members with obesity. The HEADS UP study aimed to assess the feasibility of translating a surgical and non-surgical weight loss treatment program to primary care settings across the state of Louisiana. The present report provides an overall description of the HEADS UP study methodology, study design, and procedures. This includes a review of research in support of the surgical and non-surgical weight loss procedures included in the HEADS UP study, as well as a description of the primary objectives, the screening process, and the protocols of the Intensive Medical Intervention (IMI), Intensive Medical Intervention-Translational (IMI-T) and Surgical Demonstration arms of the HEADS UP study. Screening numbers, enrollment results, and baseline participant characteristics are also presented.

## 2. Methods/Design

### 2.1. Primary Objectives of the HEADS UP Study

The primary objective of the HEADS UP study was to assess the feasibility and practicality of translating research-supported obesity treatments to a health care system. This study was particularly interested in examining the level of interest from OGB members in participating in weight loss treatment programs. HEADS UP aimed to provide a surgical and non-surgical option for weight loss treatment to OGB members that have been shown to produce significant weight loss and subsequent reduction in the burden of obesity comorbidities. The study was approved and monitored by the IRB at Pennington Biomedical Research Center (Numbers 11033, 11034 and 13018). All participants gave written informed consent.

### 2.2. Study Treatment Groups

The HEADS UP Study was divided into two initial components: the Surgical Demonstration Project and the non-surgical Intensive Medical Intervention (IMI) Demonstration Project. In the third year of the project, the IMI expanded its treatment across the state within chosen primary care practices and this expansion was subsequently referred to as the Intensive Medical Intervention-Translational (IMI-T). 

### 2.3. Outcomes of Interest

The primary outcomes for the HEADS UP study were membership interests and participation in the surgical, IMI, and IMIT components of the HEADS UP study. Membership interests were measured as the number of OGB members screened at each screening stage. Participation in each component of the HEADS UP study was measured as the number of persons enrolled. Additional outcomes examined in the HEADS UP study included changes in weight, various comorbidities of obesity, psychosocial outcomes, and medical and pharmaceutical costs associated with each treatment. These additional outcomes are being analyzed for future publications.

### 2.4. Project Setting

All screening procedures and assessments included in the surgical and the IMI components of the HEADS UP study were completed at the Pennington Biomedical Research Center (PBRC), in Baton Rouge, Louisiana. PBRC also served as the coordinating center for the IMI-T component of the HEADS UP study; however, initial IMI-T clinic and follow-up visits took place in selected primary care practices located throughout the state, and the distribution of the lifestyle change program took place predominantly through a website created for the program, www.MyWellnessPal.com, utilizing background commercial software. 

### 2.5. Surgical Demonstration Project

The HEADS UP study design included the three most common types of bariatric surgery performed during the years of study enrollment: adjustable gastric banding (AGB), Roux-en-Y gastric bypass (RYGB), and sleeve gastrectomy (SG) [14]. At study initiation, only AGB and RYGB were included because, during this time, the state funding sponsor required that 75% of surgical procedures be AGB unless otherwise medically contraindicated. However, as the study progressed, AGB was eliminated in the third year of the study and SG was offered as a third option. This decision was made because, at that time, RYGB generally performed better compared to AGB, and SG appeared to be more effective in weight loss than AGB [4,15]. 

During the start-up period of the study, there were 11 Centers of Excellence for Bariatric Surgery in Louisiana. All of these centers were contacted via mail and telephone to gauge their interest in participating in the HEADS UP Surgical Demonstration Project. Site visits were conducted for those centers that were interested and deemed appropriate for inclusion. After a review and selection process by the HEADS UP Executive Committee, 4 surgeons practicing at 3 surgical sites were selected to provide the surgical procedures. Patients approved for bariatric surgery in the HEADS UP study were permitted to choose their surgeon. 

### 2.6. Intensive Medical Intervention Demonstration Project

The IMI Component of the HEADS UP Study used a phased approach beginning with a low-calorie liquid diet (LCD) for 12 to 16 weeks. The LCD was a 800–900 kcal diet with 70 g of high-quality protein delivered in a vitamin-supplemented liquid product called Health One©, made by Health and Nutrition Technologies, Inc. PBRC has used Health One in the past and other studies using Health One have demonstrated excellent weight loss outcomes [16]. The purpose of the LCD is to achieve maximum initial weight loss. The LCD phase was then followed by a structured diet program including meal replacements and behavioral counseling. 

Behavioral change counseling was initiated during the structured diet phase and was provided to participants in a group setting. Counseling sessions were provided weekly and became gradually less frequent through the 9 months of group sessions. Group sessions began as in-person groups, and then transitioned to all sessions being conducted online in real time. Online sessions were accessed by participants using a website developed by PBRC called MyWellnessPal.com, which utilized background commercial software and made the participant experience seamless and easy. Experts in nutrition, physical activity, and behavior modification presented participants with self-management skills designed to help them engage in healthy eating, enhance their physical activity, and prevent weight regain. The IMI participants also spent time discussing strategies to stay motivated during weight loss maintenance. 

Ongoing medical management was provided using a toolbox approach. The toolbox for weight maintenance included alternative dietary approaches such as a low carbohydrate diet and various physical activity protocols. This was employed particularly for participants who were struggling with the LCD or weight loss maintenance phases of the intervention. Intervention teams were able to use the toolbox for any individual during any time of the IMI. The toolbox approach allowed for extra attention and supplemental procedures for individuals having difficulty achieving success with the IMI component. Medical staff also assisted participants on medications for chronic diseases such as diabetes, high blood pressure, and hyperlipidemia in order to safely begin and complete the LCD by making adjustments to medications as needed, and by monitoring for adverse events. 

### 2.7. Intensive Medical Intervention Translational

The IMIT component of the HEADS UP study involved the translation of the IMI program to primary care settings in Louisiana. The purpose of this component was to determine whether similar feasibility and weight changes would be observed when combining medical supervision by primary care physicians with web-based lifestyle change groups delivered by PBRC interventionists. A prior randomized control study (LOSS) sponsored by the OGB utilized the IMI design included in the current study and demonstrated that primary physicians can treat extreme obesity in the medical office [16]. LOSS showed that a significant subset of participants can achieve meaningful health benefits with medical therapy. The results of LOSS showed that 31% of participants who had lost a clinically significant amount of weight during the LCD phase managed to maintain at least 5% weight loss at 2 years follow-up.

PBRC served as the coordinating center for the study for the IMIT. Physicians and medical clinics across Louisiana were contacted to gauge their interest in treating obesity utilizing the model that we developed in the HEADS UP IMI Demonstration Project. Through a process of networking and contacting new sites, as well as communicating with physicians and clinics who have previously worked with PBRC, 12 clinics were identified to potentially serve as IMIT sites. These were a mix of private medical practices, established research centers, and academic sites located across the state. Following 9 onsite visits to determine suitability for conducting the study, 7 clinics were approved for participation by the HEADS UP Executive Committee. One site later dropped out due to staffing difficulties. 

These medical clinics acted as satellite clinics for the study and conducted all follow-up visits for IMIT participants. Primary care practitioners at the designated satellite sites were trained by PBRC to medically supervise the weight loss of participants. Participants interested in the IMIT program were referred to their nearest satellite clinic and received a similar evaluation and screening as in the IMI arm. All 2 week and annual follow-up visits were conducted at the satellite clinics. As participants were enrolled, they were placed into the lifestyle change groups conducted through MyWellnessPal.com. 

The IMIT was designed to mimic real world medical practices, so the physicians’ offices were able to manage the weight loss of participants pragmatically. As a result, physicians were able to reinstitute treatments after lapse. These included tools such as chronic disease management or reinstating the LCD and structured diet phases to assist in weight loss and weight maintenance. 

### 2.8. Screening Procedures

The study was advertised to more than 200,000 members of the Louisiana OGB Health insurance program through a mailed announcement letter and email solicitations between January and May 2012. Figure 1 shows the multi-stage screening process for the HEADS UP study. Members who were interested were directed to an OGB member website that included an information page and the initial web screening. More than 6800 members were screened through the internet website. Members were required to enter their OGB member plan as well as basic health information. OGB members were then required to watch two informational videos describing the bariatric surgery and IMI components of the HEADS UP study. 

Individuals who passed the initial web screening were subsequently contacted for a telephone interview. In addition to screening questions, these individuals were asked for their preference between the surgical or non-surgical component of the HEADS UP study. Individuals who passed the telephone screening were included into the HEADS UP lottery system. This included 1972 individuals who requested the IMI treatment arm, and 2623 individuals who requested the surgical treatment arm. Each year, the HEADS UP study design allowed PBRC to screen potential participants until 100 surgical participants and 200 participants for non-surgical treatment were enrolled. 

Various anthropometric, medical, and psychosocial information were collected during the in-person screening. In order to be eligible for the IMI or IMIT program, individuals were required to have a BMI ≥33 kg/m^2^. For the surgical component, individuals were required to have a BMI ≥40 or BMI ≥35 kg/m^2^ and type 2 diabetes. All potential enrollees were assigned the LCD and scheduled to return in 2–3 weeks to measure compliance by requiring each participant to have lost a minimum of 4 pounds from their starting weight. Each individual case was reviewed by a team of weight management experts to determine their suitability for the HEADS UP study based on the information collected during the in-person screening. 

Individuals interested in bariatric surgery were subject to additional screening procedures including dietary and psychosocial interviews examining the individuals’ understanding of bariatric surgery and their willingness and capability of implementing dietary changes, behavioral changes, and exercise prior to undergoing bariatric surgery. These individuals were further assessed by a surgical review panel consisting of physicians, surgeons, and psychologists. The decision to approve or deny participants for bariatric surgery was based on factors such as medical history, expected compliance, and psychosocial health and followed guidelines established by the American Association of Clinical Endocrinologists and the American College of Endocrinology. Participants who were approved by the surgery review panel for bariatric surgery were permitted to pick one of the four surgeons approved by the HEADS UP Executive Committee. At the conclusion of the enrollment period for the HEADS UP study, 597 participants were enrolled into the IMI program, 325 participants were enrolled into the IMIT program, and 490 participants underwent bariatric surgery. Among the surgery patients, 172 underwent AGB, 228 underwent SG, and 90 underwent RYGB.

### 2.9. Statistical Analysis

Baseline means, standard deviations, percentages, and significant correlations among participants with different groups of surgery, IMI and IMIT were calculated and are presented in Table 1. Major exclusions in the bariatric surgery and IMI programs are shown in Table 2. The schedule of study visits is summarized in Table 3. All statistical analyses were performed by IBM SPSS Statistics for Windows, version 24.0 (IBM Corp., Armonk, New York, USA).

## 3. Results 

A total of 1412 participants (597 participants at the IMI program, 325 participants at the IMIT program, and 490 participants at bariatric surgery) participated in the HEADS UP study. Table 1 contains the baseline characteristics of 1412 participants enrolled in the HEADS UP study. The majority of participants who enrolled in the study were female (84.4%) and White (60.9%). Average age of participants was 48.4 (SD 9.8) years with a range between 17 and 75. 

Participants who underwent bariatric surgery had an average weight of 131 kg (SD 19.4) and a BMI of 47.7 kg/m^2^ (SD 5.5). This is compared to 119 kg (SD 21.3) and 43.0 (SD 6.4) kg/m^2^ for IMI participants and 116 kg (SD 20.5) and 42.3 kg/m^2^ (SD 6.1) for IMIT participants. Results for comorbidity biomarkers were generally consistent among surgical, IMI, and IMIT participants. Participants were on average at high risk of type 2 diabetes with HbA1c levels of 6.1% (SD 1.2) and glucose levels of 109 mg/dL (SD 38.1). Additionally, participants were within the “prehypertensive,” range with an average systolic blood pressure of 128 mm/Hg (SD 14.0) and diastolic blood pressure of 81.0 mm/Hg (SD 8.6). LDL, HDL, and triglycerides were within normal ranges (112, 53.5, and 133 mg/dL, respectively). 

Table 2 indicates that the most frequent reasons for excluding interested participants were: (1) participant unable to comply with program procedures, e.g., could not complete screening; (2) participant was no longer interested; (3) we could not contact participant with address information provided; (4) medical exclusion; (5) did not meet the body mass index criteria.

## 4. Discussion

The HEADS UP study included a surgical component, a behavioral, lifestyle change component, and a translational component, applying the lifestyle change program within primary care settings across Louisiana. The present report provided a description of the overall HEADS UP methodology, study design and procedures.

Despite the growing observational and clinical research on weight loss treatments for obesity including bariatric surgery procedures and non-surgical, behavioral treatments, few studies have assessed the feasibility of translating this research to a health care setting. The HEADS UP study aimed to examine interest and participation in a state-insurance funded weight loss program among individuals covered by the Louisiana OGB health insurance plan. Membership interest in weight loss treatments covered by the Louisiana OGB appeared to be high given the number of individuals (6806) who elected to participate in the initial web screening for the HEADS UP study. If we estimate that half of the 200,000 covered lives at OGB are adults and we consider a reported prevalence of 7.7% for severe obesity (7.7% of 100,000 = 7700), we appear to have attracted the interest of 76% of the eligible OGB members. Following phone and in-person screenings, 490 participants elected bariatric surgery and 922 participants elected a non-surgical treatment. Of the 922 participants interested in a behavioral, lifestyle change program, 325 of them were enrolled into the IMIT component of the HEADS UP study where participants were medically managed at six satellite clinics around the state, most through primary care settings.

In addition to providing obesity treatments to more than 1400 individuals, several strengths of the HEADS UP study should be considered. First, the HEADS UP study was able to recruit and enroll a significant percentage of non-White participants (31.2% Black) for bariatric surgery. A recent review showed that affordability was an important factor in the racial disparities in the utilization of bariatric surgery [17]. The percentage of non-White participants enrolled in the HEADS UP study is significantly greater than other studies (13.8% non-White and 10.5% Black in LABS and 26% non-White in STAMPEDE) [3,10]. This demonstrates interest and willingness from minority populations to participate in obesity treatment programs. The large percentage of non-White participation was likely due to the HEADS UP study design involving a partnership with the Louisiana OGB to provide participants with a no-cost surgical or non-surgical treatment. This enabled participants to select either treatment type without considering affordability. Second, the HEADS UP study was designed to collect baseline and long-term data on medical and psychosocial factors. A relevant multi-disciplinary workshop supported by the National Institute of Diabetes and Digestive and Kidney Diseases and the National Heart Lung and Blood Institute [18] cited a lack of research focusing on predictors of bariatric surgery outcomes. Data gathered from the bariatric surgery component of the HEADS UP study will be utilized to examine the effect of baseline and change in medical and psychosocial factors on bariatric surgery success. Findings from this research will be valuable for healthcare practitioners in determining patient suitability for bariatric surgery. Furthermore, this workshop found a lack of long-term research assessing the effect of bariatric surgery on medical and pharmaceutical costs. Health insurance providers have historically been reluctant to reimburse the costs for bariatric surgery. This is despite consistent findings from observational and randomized controlled studies showing significant weight loss and improvements in type 2 diabetes [5], hypertension [4,8,13], and hyperlipidemia [4,8,13] following bariatric surgery. Through its partnership with the Louisiana OGB, the HEADS UP study will be able to assess long-term changes in patient medical and pharmaceutical expenditures following bariatric surgery. 

The primary limitation of this study is that, due to funding constraints and other requirements of the study sponsor, the treatment arms could not randomized between participants, the surgical sites or the primary care clinics. Thus, the study is a demonstration project only, and causal inferences cannot be clarified regarding study outcomes.

## 5. Conclusions

The present report provides a description of the overall HEADS UP methodology, study design, and procedures, and also presents the baseline characteristics of the data. In the future, information gathered in this study will play an important role in determining the cost effectiveness of bariatric surgery and behavioral, lifestyle change treatments for obesity. 

## Figures and Tables

**Figure 1 ijerph-17-02999-f001:**
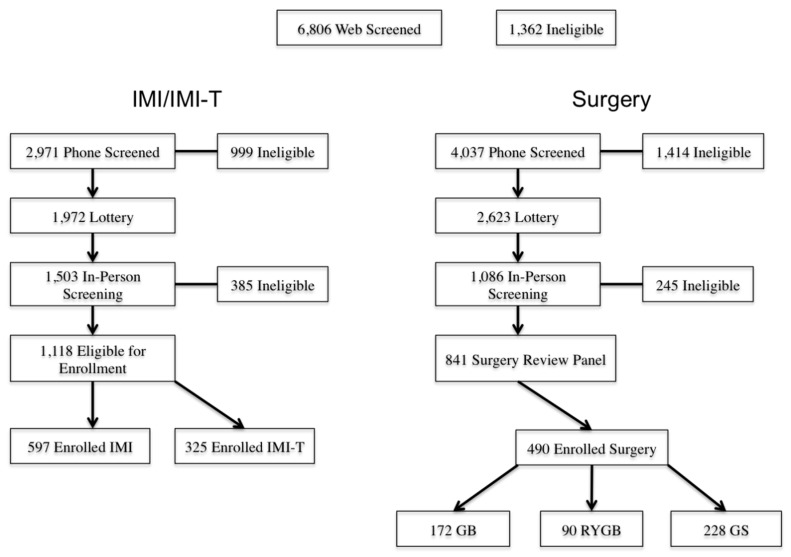
Screening Process.

**Table 1 ijerph-17-02999-t001:** Descriptive characteristics of enrolled population.

	Total (1412)	Surgery (*n* = 490)	IMI (*n* = 597)	IMI-T (*n* = 325)
Female, %	84.4	88.2	82.2	82.8
Black, %	37.6	31.6	43.6	35.7
Age, years	48.4 (9.8)	45.9 (9.4)	48.6 (9.9)	51.9 (9.1)
Weight, kg	122 (21.4)	131 (19.4)	119 (21.3)	116 (20.5)
Body mass index, kg/m^2^	44.5 (6.5)	47.7 (5.5)	43.0(6.4)	42.3 (6.1)
Systolic Blood Pressure, mm Hg	128 (14.0)	127 (13.5)	126 (14.1)	131 (13.8)
Diastolic Blood Pressure, mm Hg	81.0 (8.6)	81.0 (8.7)	81.2 (9.1)	80.9 (7.6)
HbA1c, %	6.1 (1.2)	6.2 (1.2)	6.0 (1.1)	6.2 (1.2)
Fasting glucose, mg/dL	109 (38.1)	112 (37.4)	106 (34.2)	107 (46.8)
LDL cholesterol, mg/dL	112 (32.2)	115 (32.6)	113 (31.4)	107 (33.1)
HDL cholesterol (SD)	53.5 (13.2)	53.1 (12.2)	53.4 (13.1)	54.6 (15.3)
Triglycerides (SD)	133 (77.7)	137 (79.7)	129 (74.8)	133 (80.2)

Data are the mean (SD) or percentage; LDL cholesterol, low-density lipoprotein cholesterol; HDL cholesterol, high-density lipoprotein cholesterol; IMT, Intensive Medical Intervention; IMI-T, Intensive Medical Intervention-Translation.

**Table 2 ijerph-17-02999-t002:** Screening ineligibility reasons.

	Surgery (*n* = 1384)	IMI (*n* = 1659)
Not able to comply with program procedures, %	29.1	29.4
No longer interested, %	17.5	24.9
Cannot contact	11.2	16.4
Medical exclusion	17.3	10.3
Body mass index criteria	10.9	8.9
Unable to tolerate liquid diet	0	9.2
No longer an OGB member	6.3	1.7
Unwilling to accept any surgery type	5.5	0
Psychiatric mental disorder exclusion	2.2	0

**Table 3 ijerph-17-02999-t003:** Participant schedule of study visits.

Clinic Visits	Screening	Study Visits (Week)	Follow-Up Visits (Year)
Visits	1 (Day 0)	2 (Week 2)	4	6	8	10	12	14	16	Month 6	1	2	3	Y4	5
**Intensive Medical Intervention**
Informed Consent	X														
Questionnaire	X									X	X	X	X	X	X
Height	X														
Weight	X	X	X	X	X	X	X	X	X	X	X	X	X	X	X
Waist Circumference	X	X								X	X	X	X	X	X
Inclusion/Exclusion	X	X													
Fasting Bloods	X									X	X	X	X	X	X
Lab Measurements	X									X	X	X	X	X	X
Brief Physical Exam	X										X	X	X	X	X
Medication Management	X	X	X	X	X	X	X	X	X	X	X	X	X	X	X
**Surgical Program**	Surgery						
Informed Consent	X														
Questionnaire	X									X	X	X	X	X	X
Height	X														
Weight	X	X								X	X	X	X	X	X
Waist Circumference	X	X								X	X	X	X	X	X
Inclusion/Exclusion	X	X													
Fasting Bloods										X	X	X	X	X	X
Lab Measurements	X									X	X	X	X	X	X
Brief Physical Exam	X									X	X	X	X	X	X
Medication Management	X									X	X	X	X	X

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
