# Peer review of "HEADS UP: Design and Methods of a Louisiana State-Funded Surgical and Non-Surgical Weight Loss Program"

_ijerph, 2020, doi:10.3390/ijerph17092999_

Round 1

Reviewer 1 Report

Major comments

The current paper describes a recruitment project for weight loss program. The paper is of very limited implication due to two main reasons. Firstly, it describes a specific setting and health system that most probably cannot be applicable in many others in the US or any other place. Secondly, the non-availability of clinical outcomes in terms of weight-loss rate and maintenance limits the study importance. As a clinician and researcher in the field of obesity treatment I found the study of limited utility to be published in an international journal.

Other comments

  • The title needs to be clarified
  • According to the journal guideline, the abstract should figure as 1 paragraph with no subsection and that does not exceed 200 words.
  • The introduction is long and redundant need to be shortened not to exceed 700-800 words
  • The authors conducted some statistical analysis, however no description of those has been reported under a statistical analysis subsection in the Method section.
  • The discussion is a pure speculation, should be structured differently in the following order: (i) The main finding of the current study and comparison with similar studies already published; (ii) The clinical implication; (iii) The strengths and limitations and (iv) New direction of future results. 

Author Response

Responses to Reviewer 1’s comments:

Major comments

The current paper describes a recruitment project for weight loss program. The paper is of very limited implication due to two main reasons. Firstly, it describes a specific setting and health system that most probably cannot be applicable in many others in the US or any other place. Secondly, the non-availability of clinical outcomes in terms of weight-loss rate and maintenance limits the study importance. As a clinician and researcher in the field of obesity treatment I found the study of limited utility to be published in an international journal.

Response: Thanks for your comments. The present report intended to provide an overall description of the HEADS UP methodology, study design, and procedures. Preliminary outcomes have been reported previously, and we are still collecting outcomes and will present the final results soon.

Other comments

The title needs to be clarified

According to the journal guideline, the abstract should figure as 1 paragraph with no subsection and that does not exceed 200 words.

The introduction is long and redundant need to be shortened not to exceed 700-800 words

The authors conducted some statistical analysis, however no description of those has been reported under a statistical analysis subsection in the Method section.

The discussion is a pure speculation, should be structured differently in the following order: (i) The main finding of the current study and comparison with similar studies already published; (ii) The clinical implication; (iii) The strengths and limitations and (iv) New direction of future results.

Responses: Thanks for your comments. We have re-edited the manuscript according to your suggestions: modify the title, re-edit the abstract, shorten the Introduction Section, add a Statistical Section, and re-edit the Discussion Section.

Reviewer 2 Report

Dear Authors this is an interesting paper but i have some criticisms:

  • first  please re.write all sections; it is very difficult to follow the manuscript. i.e.: into the introduction there are data about materials and methods , and during materials and methods there is a mix between description and discussion.
  • please use a scientific division.
  • i.e,. section 2.4 line 16: what do you mean with...4 surgeons at 3 surgical sites? this is confused

your data are important and in my opinion the paper shoudl be focused of description. why you report data about surgical literature review into the results and reported that you change the gastric band after 3 years? so you have also data about weight loss?

Author Response

Responses to Reviewer 2’s comments:

Dear Authors this is an interesting paper but i have some criticisms:

First, please rewrite all sections; it is very difficult to follow the manuscript. i.e.: in the introduction, there are data about materials and methods, and during materials and methods there is a mix between description and discussion.

Please use a scientific division.

Response: Thanks for your comments. We have re-edited the manuscript.

Section 2.4 line 16: what do you mean with...4 surgeons at 3 surgical sites? this is confused your data are important and in my opinion the paper should be focused of description. why you report data about surgical literature review into the results and reported that you change the gastric band after 3 years? so you have also data about weight loss?

Response: We have re-edited the Method and Result Sections according to your suggestions. Based on the requirements of our funding sponsor, and updated literature on the topic, after 3 years we have changed the gastric band procedure to sleeve gastrectomy. We have moved this part to the Method Section. The present report is intended to provide an overall description of the HEADS UP methodology, study design, and procedures. Preliminary outcomes have been reported previously, and we are still collecting outcomes and will present the final results soon.

“4 surgeons at 3 surgical sites??” means 2 surgeons practicing at 1 surgical site.

Reviewer 3 Report

This observational study aims to examine the feasibility of a state-funded weight loss program through two weight loss approaches: a lifestyle intervention or bariatric surgery. The study is very important to control obesity and its related complications worldwide

  • The abstract is well-written and clear.
  • Introduction: the importance of the study has to be more detailed.
  • The causes of ineligibility are not clearly described
  • Results: The authors only analyzed the baseline characteristics of the study participants. I suggest the correlation between the independent variables will provide more significant findings
  • Analysis of the data is lacked for more details
  • Provide bar charts for your statistical reports to improve the quality and readability of the paper.
  • The authors provide that (information gathered in this study will play an important role in determining the cost-effectiveness of bariatric surgery and behavioral, lifestyle change treatments for obesity). Regarding that, what is the conclusion of the study?

The study is very important and very rich by data collected within 5 years of observation, while these data require more analysis and the discussion has to support the main finding of this study.

Author Response

Responses to Reviewer 3’s comments:

This observational study aims to examine the feasibility of a state-funded weight loss program through two weight loss approaches: a lifestyle intervention or bariatric surgery. The study is very important to control obesity and its related complications worldwide

The abstract is well-written and clear.

Introduction: the importance of the study has to be more detailed.

The causes of ineligibility are not clearly described

Results: The authors only analyzed the baseline characteristics of the study participants. I suggest the correlation between the independent variables will provide more significant findings

Analysis of the data is lacked for more details

Provide bar charts for your statistical reports to improve the quality and readability of the paper.

Response: Thank you so much for your comments. Per your suggestions we have added more details in the introduction, the causes of ineligibility, on statistical analyses and study samples, and on the baseline characteristics of the study sample.

The authors provide that (information gathered in this study will play an important role in determining the cost-effectiveness of bariatric surgery and behavioral, lifestyle change treatments for obesity). Regarding that, what is the conclusion of the study?

The study is very important and very rich by data collected within 5 years of observation, while these data require more analysis and the discussion has to support the main finding of this study.

Response: Thanks for your positive comments. We have added more details in the Conclusions. We are still collecting outcome data and will present the final results soon.

Reviewer 4 Report

The authors highlight design and methods which is important for other researchers who are performing clinical trials in this area.  

What screening tool did you use to randomize? 

What variables are you tracking overtime (body weight, % body weight loss, body fat etc)? How often are these being collected? 

Author Response

Responses to Reviewer 4’s comments:

The authors highlight design and methods which is important for other researchers who are performing clinical trials in this area. 

What screening tool did you use to randomize?

Response: We have added more details about randomization.

What variables are you tracking overtime (body weight, % body weight loss, body fat etc)? How often are these being collected?

Response: We have added one table to show follow-up visits.

Round 2

Reviewer 1 Report

Accept 

Reviewer 3 Report

No further comments